# PICALM and Alzheimer’s Disease: An Update and Perspectives

**DOI:** 10.3390/cells11243994

**Published:** 2022-12-10

**Authors:** Kunie Ando, Siranjeevi Nagaraj, Fahri Küçükali, Marie-Ange de Fisenne, Andreea-Claudia Kosa, Emilie Doeraene, Lidia Lopez Gutierrez, Jean-Pierre Brion, Karelle Leroy

**Affiliations:** 1Laboratory of Histology, Neuropathology and Neuroanatomy, Faculty of Medicine, Université Libre de Bruxelles, ULB Neuroscience Institute, 808 Route de Lennik, 1070 Brussels, Belgium; 2Complex Genetics of Alzheimer’s Disease Group, VIB Center for Molecular Neurology, VIB Antwerp, Department of Biomedical Sciences, University of Antwerp, 2000 Antwerp, Belgium

**Keywords:** *PICALM*, Alzheimer’s disease, GWAS, amyloid β, neurofibrillary tangles, microglia

## Abstract

Genome-wide association studies (GWAS) have identified the *PICALM* (Phosphatidylinositol binding clathrin-assembly protein) gene as the most significant genetic susceptibility locus after *APOE* and *BIN1*. PICALM is a clathrin-adaptor protein that plays a critical role in clathrin-mediated endocytosis and autophagy. Since the effects of genetic variants of *PICALM* as AD-susceptibility loci have been confirmed by independent genetic studies in several distinct cohorts, there has been a number of in vitro and in vivo studies attempting to elucidate the underlying mechanism by which PICALM modulates AD risk. While differential modulation of APP processing and Aβ transcytosis by PICALM has been reported, significant effects of PICALM modulation of tau pathology progression have also been evidenced in Alzheimer’s disease models. In this review, we summarize the current knowledge about PICALM, its physiological functions, genetic variants, post-translational modifications and relevance to AD pathogenesis.

## 1. Physiopathology of Alzheimer’s Disease and Genetic Risk Factors

Alzheimer’s disease (AD) is the most common cause of dementia, manifesting itself in cognitive deficits. AD is characterized by two neuropathological hallmarks: extracellular deposition of fibrils made up of amyloid β (Aβ) peptides as senile plaques [1] and intracellular accumulation of hyperphosphorylated tau as neurofibrillary tangles (NFTs) [2] (for review, see [3]).

Aβ is produced by successive cleavages of β-amyloid precursor protein (APP), encoded by *APP* gene located in chromosome 21, by β-secretase (BACE1) and γ-secretase [4]. APP is cleaved by BACE1 at its N terminal domain to release soluble APP-β and the membrane-bound APP C-terminal fragment (β-CTF or C99). The γ-secretase is a complex of four protein subunits consisting of presenilin (PSEN), Nicastrin, presenilin enhancer (PEN) and APH1 (anterior pharynx-defective 1). Sequential cleavages of APP by BACE1 and γ-secretase lead to production of Aβ peptides of 38-, 40- and 42-amino acids in length. Aβ peptides are principally produced in endosomes and are released from neurons in a synaptic activity-dependent manner [5]. Aβ peptides adopt aggregation-prone conformations that are resistant to proteolysis and form oligomers, protofibrils and fibrils. Due to hydrophobicity at its C-terminus, Aβ42 has greater aggregation properties [6].

Tau is a microtubule-associated protein and has a role in stabilizing neuronal microtubules. Tau is encoded by *MAPT* (microtubule-associated protein tau) gene located on chromosome 17. Tau is natively soluble and unfolded, but tau can be aggregated into oligomers and fibrils in the presence of pathologically misfolded tau or polyanions [7]. In AD, tau aggregates form paired helical filaments (PHF) or straight filaments [8]. Compelling evidence suggests that tau pathology, defined by the accumulation of hyperphosphorylated aggregated forms of tau, is strongly associated with neurodegeneration and cognitive impairment in AD [6,9]. Tau pathology propagates between neuroanatomically connected areas throughout the brain during the progression of AD. The stereotypical aspect of tau pathology progression forms the basis of neuropathological Braak staging of tau pathology [10]. The stage of tau pathology better correlates with cognitive decline than Aβ load in AD patients [9]. Numerous studies have suggested that both synthetic and AD brain-derived tau fibrils can propagate in a prion-like manner by recruiting normal unfolded tau into pathological tau aggregates [11]. Tau aggregation can also be found in neurons and glia of other human neurological disorders, so-called tauopathies, such as frontotemporal lobar degeneration (FTLD) with tau pathology, progressive supranuclear palsy (PSP), Pick disease, corticobasal degeneration (CBD), etc. [12].

AD can be generally classified into two subgroups according to the age of onset. AD cases occurring earlier than age 65 are termed early-onset AD (EOAD), constituting less than 5% of all cases. Early-onset AD includes familial AD (FAD) inherited in an autosomal-dominant manner and can present at a very early age of onset and with a more rapid rate of disease progression, constituting 1–2% of AD cases. A number of FAD mutations have been identified in the genes of *APP*, *Presenilin1* (*PSEN1*) and *Presenilin2* (*PSEN2*) (γ-secretase component) [13].

Since age is the most important risk factor for AD [14], the great majority of cases, estimated at more than 95%, occur after age 65, constituting late-onset AD (LOAD). There was little knowledge of genetic risk factors for LOAD except for *Apolipoprotein* (*APOE*) risk allele *APOE4* till the breakthrough made by genome-wide association studies (GWAS) [15,16]. To date, GWAS have identified about 75 single nucleotide polymorphisms (SNPs), including *APOE4*, *BIN1*, *PICALM*, *CLU*, *CR1*, *ABCA7*, *TREM2*, *MS4A* and *APH1B,* with a significant association for LOAD susceptibility [17]. These LOAD-susceptibility genes are functionally implicated in multiple cellular processes such as synaptic function, lipid metabolism, inflammation, endocytosis, cytoskeletal transports, and tau and amyloid pathways in AD pathologies [18,19,20]. Nevertheless, the mechanisms by which the GWAS hit genes modulate the AD risk remain largely elusive and have been under active scrutiny in the field of functional genomics in the post GWAS era. The vast majority of these variants are located in the noncoding region [21], and such variants are supposed to exert phenotypic effects via the perturbation of transcriptional gene promoters and enhancers [22].

Among them, several independent GWAS have identified SNPs significantly associated with LOAD in the *PICALM* gene (see Section 3.1) coding phosphatidylinositol binding clathrin-assembly protein PICALM (also known as CALM that stands for clathrin assembly lymphoid myeloid leukemia). Numerous GWAS have suggested that *PICALM* is one of the most significant susceptibility genes for LOAD after *APOE4* and *BIN1* [15,17,19,23,24,25,26,27]. Since the discovery of the genetic implication of *PICALM* in LOAD, many studies have been conducted to decipher the roles of PICALM in AD pathogenesis. In light of these findings, we describe in this review the protein structure and known post-translational modifications of PICALM. We summarize the emerging link between PICALM and pathogenic proteins found in AD brains to better understand the mechanisms by which PICALM contributes to AD pathogenesis via Aβ-dependent and Aβ-independent pathways.

## 2. Biology of PICALM

### 2.1. Tissue Expression, Cellular Functions and Protein Interaction of PICALM

The human *PICALM* gene is located on chromosome 11 [28]. There are 24 transcripts as splice variants, 15 of them coding protein [28], whose sizes vary from 134 to 652 amino acids [29], and the molecular weights of the major isoforms are estimated at 65–75 kDa due to alternative splicing of exon 13 [30]. PICALM is a ubiquitous protein and is expressed in the brain, muscle, kidney, urinary bladder, connective tissues, bone marrow, lymphoid tissues and in female tissues, including breast and placenta, https://www.proteinatlas.org/ENSG00000073921-PICALM/tissue (accessed on 1 June 2022). [31]. In the human brain, PICALM is abundantly expressed in the microglia, oligodendrocytes, endothelial cells, neurons, vascular mural and choroid plexus cells (for further information on the localization of *PICALM* in the AD brain, see 3.3.1) [30,32,33,34,35,36]. In the mouse brain, *PICALM* mRNA is highly expressed in the hippocampus [31], in particular in the dentate gyrus granule cells, as detected by a spatial transcriptomic study [37,38].

STRING database analysis revealed predicted protein–protein interactions of PICALM [39,40]. PICALM interacts with key proteins for clathrin-mediated endocytosis, intracellular trafficking and signaling (Figure 1). Interestingly, PICALM tightly interacts with other proteins encoded by GWAS hit genes for LOAD, such as *BIN1*, *CD2AP*, *EPHA1*, *AP2A1* and *AP2A2* [41]. PICALM is almost perfectly colocalized with the AP2 adaptor and AP2 α-adaptin subunits (encoded by *AP2A1* and *AP2A2*) [42,43]. Interestingly, gene variants, splicing defects and altered expression of *AP2A1* and *AP2A2* have been recently shown to have a significant association with LOAD susceptibility [41]. Mounting evidence suggests PICALM has multiple roles in cellular functions and homeostasis as a clathrin adaptor for clathrin-mediated endocytosis, erythroid maturation, transferrin uptake, lipid homeostasis, autophagy, neuronal polarity, neuritic prolongation and synaptic vesicle turnover [44].

#### 2.1.1. PICALM and Clathrin-Mediated Endocytosis

Clathrin-mediated endocytosis, also called receptor-mediated endocytosis, is a process leading to the internalization of plasma membrane-associated cargo molecules and subsequent trafficking through intracellular vesicle compartments. PICALM is necessary for clathrin assembly at the plasma membrane by binding to AP2 and clathrin (Figure 2A). PICALM interacts simultaneously with AP1 and clathrin at the trans-Golgi network in HeLa cells [43]. Several independent studies have reported that the depletion of PICALM leads to an increase in the size of clathrin-coated vesicles (CCVs) and to an increase in heterogeneity in the CCVs shape in cultured cells of rat hippocampus [46] and of HeLa cells [43,47]. CRISPR/Cas9-mediated PICALM disruption also leads to an increase in the number of early endosomes in HeLa cells [48]. A similar observation was obtained by the depletion of uncoordinated protein-11 (unc-11), the ortholog of PICALM in neurons of *C. elegans* [49]. Another study suggests that PICALM is required for efficient clathrin coat maturation in fibroblasts [50].

#### 2.1.2. PICALM, Hematopoiesis, Transferrin-Uptake and Cholesterol Homeostasis

Chromosomal translocations of *PICALM* have been observed in patients with acute lymphoblastic leukemia and acute myeloid leukemia [52]. In mice, nonsense point mutations of *PICALM* cause abnormal iron distribution, growth retardation, shortened life span and hematopoietic abnormalities [53].

PICALM is a critical protein, and homozygous *PICALM* knockout mice are dwarfed and die within 1 month after birth due to a deficiency in erythroid maturation and transferrin uptake [54]. There are controversies in the literature on the effects of PICALM on transferrin uptake. While some studies suggest that PICALM overexpression or down-expression did not alter transferrin uptake in HeLa cells or HEK293 cells [55,56], others observed a significant effect on transferrin uptake in the liver, bone marrow and erythroid cells from *PICALM*-deficient mice [50,54]. It is thus speculated that the deletion of PICALM may be partially compensated by other endocytic proteins in certain types of cells. Indeed, PICALM is dispensable in myeloid and B-lymphoid development [50].

Additionally, PICALM also modulates cellular cholesterol homeostasis, as PICALM expression influences the expression of genes encoding proteins involved in cholesterol biosynthesis and lipoprotein uptake [57]. Knockdown of PICALM leads to up to a 50% increase in cholesterol biosynthesis genes in in HEK293, murine embryonic fibroblasts (MEF) and Cath-a-differentiated (CAD) neuronal cells [57].

#### 2.1.3. The Roles of PICALM in Neuronal Polarity and Synaptic Vesicle Sorting

PICALM and its neuronal homolog assembly protein 180 (AP180) have quite interesting functions in neuronal cells. Endocytosis is reduced in PICALM- or AP180-deficient neurons [58]. Interestingly, PICALM and AP180 have critical roles in establishing the polarity and neuritic growth in neuronal cells. While treatments with siRNA targeting *PICALM* lead to dendritic dystrophy, treatments with siRNA targeting *AP180* abolish axonal extension in rat primary neuronal culture. In contrast, neurons overexpressing AP180 or PICALM generate multiple axons [58].

PICALM is expressed in neurons and is present in synapses [36]. PICALM regulates membrane fusion by binding to R-SNAREs (soluble NSF attachment protein receptors) such as VAMP2, VAMP3 and VAMP8 [59,60]. Furthermore, PICALM regulates the internalization of VAMP2 (Synaptobrevin 2), the most abundant synaptic vesicle protein and a major R-SNARE component [56]. Since the AP180 N-terminal homology (ANTH) domain (see chapter 2.2) of PICALM interacts with VAMP2, both PICALM and AP180 regulate endocytic sorting of synaptic vesicles at active zones for neurotransmission [59]. PICALM and related ANTH-domain-containing proteins modulate the surface levels of calcium-permeable AMPA-type glutamate receptors (AMPARs) that mediate fast excitatory neurotransmission and excitotoxicity [61]. PICALM is thereby supposed to play roles in synaptic plasticity and learning by modulating both long-term potentiation (LTP) and long-term depression (LTD) [62].

#### 2.1.4. PICALM and Autophagy

PICALM plays a key role in multiple steps in autophagy (Figure 2B). PICALM regulates the endocytosis of VAMP2, VAMP3 and VAMP8, which are essential for the autophagy process [60]. PICALM knockdown leads to a reduced fusion of autophagosome precursor vesicles, which is a VAMP2- and VAMP3-dependent process, and thus decreases autophagosome biogenesis. PICALM knockdown also influences the process of autophagosome-lysosome fusion, which is a VAMP8-dependent process [63]. Therefore, PICALM reduction results in a general reduction in autophagic flux in HeLa, HEK, CAD and MEF cells [51]. Additionally, PICALM is also implicated in the maturation of Cathepsin D, which is a critical lysosomal component [48]. Disruption of exon1 of *PICALM* leads to deficits in the maturation of Cathepsin D and autophagy in HeLa cells [48].

### 2.2. Protein Structure of PICALM

PICALM is constituted of an N-terminal ANTH-domain binding phosphatidyl inositol 4,5 bisphosphate (PtdIns(4,5)P_2_) and a C-terminal clathrin adaptor domain (Figure 3A) [42,64].

The ANTH domain is a membrane-binding domain with specificity for phosphatidylinositol 4,5-bisphosphate (PtdIns(4,5)P_2_) and acts as a universal adaptor for nucleation of clathrin coats. In silico modeling of N-terminal ANTH domain suggests a structure consisting of 11 α-helices [60] (Figure 3B). The PtdIns(4,5)P_2_ binding residues (Kx9Kx(K/R)(H/Y) are present on several helices. The PtdIns(4,5)P_2_ binding sites contain evolutionally conserved basic amino acid residues (K28, K38, K40 and K41, numbering according to the PICALM isoform of 652 amino acids) forming a positively charged patch on the surface to interact with phosphates of PtdIns(4,5)P_2_ [64]. The ANTH domain also contains the binding region for R-SNARE proteins, and PICALM can bind simultaneously PtdIns(4,5)P_2_ and VAMP8 [60]. PICALM interacts with R-SNARE proteins via its hydrophobic amino acid residues such as L219, F240, M244, I272 and L251 [60]. A recent study has shown that the ANTH domain directly binds to ubiquitin, and the ANTH domain can act as an adaptor for ubiquitinated protein [67].

In the C-terminal clathrin adaptor domain of PICALM contains critical motifs that allow PICALM to interact with clathrin and other endocytic machinery proteins. The clathrin binding motif is DLLDLQ (392–397 in the isoform of 652 amino acids). AP2-binding motifs are DIF (D375, I376 and F377), DPF (D420, P421 and F422) and FESVF (F489, E490, S491, V492 and F493). The asparagine-proline-phenylalanine (NPF) motif binds to Eps15 homology (EH) domains found in proteins associated with endocytosis and vesicle trafficking. Since exon 13 contains one DPF (AP2 binding) and one NPF (EH domain binding) motif, the splice variant without exon 13 is supposed to have reduced interaction with AP2 and other protein partners owning EH-domains [43].

### 2.3. Post-Translational Modifications of PICALM

PICALM undergoes several post-translational modifications (PTM), such as phosphorylation, methylation, ubiquitination, sumoylation and acetylation [65,66,68,69,70,71,72,73,74]. These post-translational modifications have been identified on numerous amino acid residues of PICALM: 41 phosphorylation, 15 ubiquitination, 1 acetylation, 1 mono-methylation, 1 di-methylation and 1 sumoylation sites have been described so far. Certain factors are known to induce some of the post-translational modifications on PICALM, such as phosphorylation by Aβ treatment in primary neuronal culture [75], by BI2536, a selective inhibitor of protein kinase Polo-Like Kinase 1 (PLK1), inducing phosphorylation (Y138, T146 and S537) [76], by ischemia (S303 and S315) [77] and by nocodazole, an inhibitor of microtubule polymerization (S474 and S645) [78]. The most well-characterized PTM of PICALM are the four phosphorylation sites (S16, Y96, S107 and S303) and two ubiquitination sites (K318 and K324) that have been validated by mass-spectrometry and have been described for more than 5 times [65]. While the phosphorylation on the ANTH domain may influence the interactions of PICALM with PtdIns(4,5)P_2_ or R-SNARE proteins, the other phosphorylation residues are located in the C-terminal clathrin adaptor domain exhibiting pivotal functions in protein–protein interaction of PICALM with clathrin and other endocytic machinery proteins. Therefore, it is speculated that phosphorylation of PICALM should play a role in regulating interaction with PtdIns(4,5)P_2_ and other protein binding partners. For instance, the R-SNARE binding domain contains several sites that can undergo PTM, such as one phosphorylation (Y237), two ubiquitination (K231 and K252) and one sumoylation (K238), and thus, these PTM may regulate the interaction between PICALM and R-SNARE.

PICALM is a substrate of calpain and caspase [33,79,80]. Nevertheless, it remains largely unknown if the phosphorylation and ubiquitination of PICALM affect its protein stability or degradation.

Given that PICALM contains more than 130 putative phosphorylation sites, PICALM phosphorylation may be dysregulated in AD brains not only in terms of its global phosphorylation level but also in the pattern of site-specific phosphorylation. Indeed, Aβ exposure increases PICALM phosphorylation in primary rat neuronal cortical culture [75] though phosphorylation sites and the involved kinases remain to be identified. While it is speculated that phosphorylation may change the interaction of PICALM with its binding partners such as R-SNAREs, PtdIns(4,5)P_2_ or clathrin, it remains elusive which kinases are involved in PICALM phosphorylation, whether PICALM phosphorylation is misregulated during the progression of AD, whether PICALM phosphorylation has any impact on its protein stability/degradation or its subcellular localization and whether phosphorylation of PICALM may cause a loss of function and/or a gain of toxicity.

## 3. PICALM and Alzheimer’s Disease Pathology

### 3.1. PICALM Gene and AD

#### 3.1.1. Gene Location and Genetic Variants of *PICALM*

In humans, the *PICALM* gene is located on chromosome 11q14 (Figure 4). Since several independent GWAS have reported the involvement of *PICALM* SNPs in LOAD susceptibility, many other studies have replicated and confirmed the association of *PICALM* genetic variants with LOAD in diverse cohorts (see review [44]). There have been several LOAD-associated SNPs identified within or near the *PICALM* gene so far (Appendix A). A recent large GWAS (*n* = 788,989) suggests that the most significant SNP associated with LOAD susceptibility is rs3851179, located in the intergenic regions between *PICALM* and *EED* [17].

rs3851179^T^ is a protective allele reported to reduce AD risk by 9–29% among various cohorts [81]. rs3851179 is located in a region 88 kb upstream of the *PICALM* gene (Figure 4) and is likely involved in transcription factor binding [84].

rs3851179 is in linkage disequilibrium (LD) with other genome-wide significant SNPs associated with LOAD, such as rs10792832 [24], rs1237999 [81], rs561655 [82] and rs541458 [15] (r^2^ = 1, r^2^ = 0.7, r^2^= 0.762 and r^2^ = 0.622, respectively) (Figure 5; Appendix A).

There are no LOAD-associated coding variants of *PICALM* [29]. Two rare nonsynonymous coding variants in *PICALM* (rs147556602 (p.P495A) and rs117411388 (p.H458R)) have been reported, although their association with AD remains unclear [88]. Computational analyses suggest that rs561655 and rs573167, located in the intergenic region, have significant deleterious effects based on combined annotation-dependent depletion (CADD) scores (CADD 17.79 for rs561655 and 15.63 for rs573167) [89].

Furthermore, two *PICALM* SNPs, not genome-wide significant, are known to be involved in splicing regulation (Figure 4). rs592297, located in exon 5, is an exonic splice enhancer and is associated with *PICALM* lacking exons 2–4 [30]. rs588076, located in *PICALM* intron 17, is associated with allelic expression imbalance (AEI) of the *PICALM* isoform lacking exon 18-19 [83].

#### 3.1.2. *PICALM* and Quantitative Trait Loci (QTL)

Analyses on expression quantitative trait loci (eQTL) suggest that there may be a cell-type specific effect of the *PICALM* variant, notably in microglia. Several recent *PICALM* eQTL analyses have consistently suggested that there is a significant colocalization between AD GWAS signals and *PICALM* eQTL in microglia [90,91,92,93]. *PICALM* rs10792832 (in almost full LD with rs3851179) is located in an open chromatin region (OCR), and the non-protective variant was associated with both lower OCR signal and gene expression by regulating chromatin accessibility [91].

Microglia may not be the only cell type affected by *PICALM* variants. The protective rs3851179^T^ allele is linked to increased *PICALM* expression in the brain [81,94], especially implicated in *PICALM* expression in endothelial cells [30,34]. The rs3851179^T^ allele is associated with increased *PICALM* mRNA and protein expression in induced pluripotent stem cells (iPSC) differentiated into endothelium compared to isogenic cells bearing non-protective rs3851179^C^ [34]. The non-protective rs3851179^C^ allele is associated with a significant reduction in *PICALM* isoform 2 mRNA expression in the frontal and in temporal cortex [94].

Alternative splicing of *PICALM* is significantly changed in AD brains in relation to LOAD-susceptibility variants [41]. Differential excision of introns in the *PICALM* gene was observed in the dorsolateral prefrontal cortex (DLPFC) of AD brains in correlation with the GWAS signal [41]. Raj and colleagues observed such alteration of alternative splicing in iPSC-derived neurons overexpressing tau, suggesting that accumulation of tau drives abnormal alternative splicing of key AD genes, including *PICALM*.

It has to be noted that the nearest coding gene of rs3851179 is *EED* [17]. A colocalization of the *EED* function association has been reported [95]. It remains elusive if *EED* is also involved in AD pathogenesis.

These data support the hypothesis that *PICALM* is the risk gene in the locus and that the potential effect for AD risk is, at least partially, via modulation of microglial *PICALM* expression.

#### 3.1.3. *PICALM* Variants and Phenotypes Relevant to AD

Studies have shown significant associations of some of the SNPs in the *PICALM* locus with phenotypes relevant to AD, such as the age of onset, cognitive functions, information processing speed, spontaneous brain activity, hippocampal atrophy and A*β* or tau levels in cerebrospinal fluid (CSF) (Table 1). Some SNPs are not genome-wide significant to LOAD (rs510566, rs592297, rs17148741) but showed statistically significant association with hippocampal atrophy or A*β* and tau levels in cerebrospinal fluid (CSF) [81,96].

*PICALM* rs541458^T^ risk allele is associated with decreased Aβ in CSF [98]. A potential association of rs541458 was reported with the spontaneous brain activity of amnestic MCI [99] and with the cognitive performance of non-demented Chinese in an age-dependent manner [109].

Xu et al., reported that rs1237999, located 34 kb upstream of *PICALM*, is localized in a CCCTC binding factor in a regulatory region (Figure 4). rs1237999 in LD with rs3851179 and is associated with the levels of Aβ, tau and the ratio of phosphotau/Aβ in CSF [81].

Interestingly, rs10501610 is not in LD with rs3851179 (r^2^ = 0.082), but genome-wide significant (*p* = 1.186 × 10^−12^) and is associated with CSF biomarkers [81]. rs2888903, rs7941541 and rs10751134 are associated with the age of onset or cognitive function [101]. rs7110631 in LD (r^2^ = 0.740) is associated with age-dependent cognitive decline [102].

There is a significant association of *PICALM* rs3851179^T^ protective allele with the change rate of adjusted hippocampal volume [103,104] and entorhinal cortex thickness [103,108] measured by magnetic resonance imaging (MRI). rs3851179^T^ is associated with better cognitive functions in the oldest old [106], metabolic connectivity [110], and default mode network function in mild cognitive impairment (MCI) [111]. Non-protective rs3851179^C^ allele is associated with the decline in the information processing speed [112] and earlier onset of AD [105].

It has to be noted that some studies have a rather small sample size (Table 1) and may be underpowered. Further replication of these studies in larger cohorts is warranted. Although correlation does not imply causation, these reports support the hypothesis that *PICALM* variants are associated with phenotypes relevant to AD.

### 3.2. PICALM and Aβ Pathology

#### 3.2.1. PICALM and APP Processing

Given that the generation of Aβ peptides from APP and its accumulation are pivotal events in AD pathogenesis, researchers have sought to identify the role of PICALM on Aβ pathology. Multiple lines of evidence suggest that PICALM regulates APP processing [35,113,114,115] and Aβ transcytosis [34].

Tian et al., have reported a rather protective role of PICALM in APP metabolism. PICALM forms a complex with APP-CTF together with AP2 and microtubule-associated protein 1 light chain 3 (LC3), an autophagy marker [115]. The interaction of APP-CTFs, PICALM, AP2 and LC3 results in transporting APP-CTFs into the autophagosome in which APP-CTFs are degraded by autophagy. The interaction between APP-CTFs and PICALM consequently leads to a reduction of Aβ production [114]. This critical role of PICALM in autophagy is strengthened by another study showing that loss of exon 1 of PICALM in HeLa cells leads to an augmentation of lysosomal enzymes in the endosome-enriched fraction and accumulation of autophagy substrates such as APP-CTFs [48].

PICALM regulates APP processing via γ-secretase internalization due to a direct interaction between the PICALM ANTH domain and Nicastrin, one of the components of the γ-secretase complex [116]. Intriguingly, PICALM itself modulates APP-CTF production via controlling BACE1 expression [94].

In addition, PICALM has been reported to be a regulator of PICALM/clathrin-dependent internalization of Aβ bound to the low-density lipoprotein-receptor-related protein-1 (LRP1), a major Aβ receptor, leading to Aβ transcytosis and clearance [34].

As there are multiple implications of PICALM in the regulation of Aβ pathology (autophagy, APP processing, transcytosis), it is not surprising that there are some contradictory observations in transgenic models of amyloid pathology with reduced expression of PICALM. One study has shown a significant increase in amyloid load in PICALM^+/−^ crossed with APP^sw/0^ and observed a 3.5-4-fold increase in Aβ load in the 9-month-old APP^sw/0^PICALM^+/−^ mouse brain [34]. On the other hand, Kanatsu et al., observed a significant reduction of Aβ load in 12-month-old A7/PICALM^+/−^ mouse brains. Further, Xiao et al., have reported that PICALM is colocalized with APP and PICALM expression level has a significant impact on Aβ level in APP/PS1 mice injected with adeno-associated virus (AAV)-mediated modulation of PICALM levels. While PICALM overexpression increases APP internalization and Aβ production, PICALM knockdown decreases amyloid load [35]. Furthermore, drosophila PICALM orthologue LAP overexpression did not alter the Aβ42 level in a drosophila model overexpressing Aβ42 [117]. The discrepancy of the observation on Aβ production may be related to the different transgenes with distinct mutations and the differences in amyloid doses in the models analyzed. These contradictory data imply that the effect of PICALM reduction can differ depending on the progression of Aβ pathology. Further studies are necessary to determine the function of PICALM in APP processing by using a model close to the physiological conditions of human LOAD to achieve clinical relevance.

#### 3.2.2. *PICALM* and Aβ-Mediated Neurotoxicity via Glutaminergic Neurotransmission

In the yeast model, PICALM orthologue was identified as a protective modifier of Aβ toxicity [118]. Overexpression of YAP1802, the yeast orthologue of PICALM, partially rescues Aβ-induced toxicity without affecting the secretory pathway. Treusch et al., confirmed the rescuing effect of PICALM in *C. elegans* and rat cortical neuronal culture: overexpression of unc-11, the *C. elegans* orthologue of PICALM, rescues Aβ-induced glutaminergic neuron loss in *C. elegans* and lentiviral PICALM overexpression diminished Aβ-induced neuron loss in primary culture of rat cortical neurons. On the contrary, D’Angelo and colleagues observed that mammalian PICALM enhanced Aβ-induced toxicity in a yeast model overexpressing Aβ tagged with GFP [119]. These inconsistent observations are partly due to the heterogeneity of the Aβ species and the distinct expression level of Aβ. In addition, Aβ overexpression in these bio-engineered yeast models was driven by transgene encoding Aβ, and thus, the nature of overexpressed Aβ should differ from the Aβ peptides derived from successive cleavages of APP in a physiological/pathophysiological condition in human brains.

It is noteworthy that Aβ is not co-immunoprecipitated with PICALM in human brains, and PICALM immunoreactivity is never observed in the center of amyloid plaques [32,33,34,35]. Thus, PICALM may presumably play roles in Aβ-induced toxicity in an indirect manner. One of the candidate intermediates between PICALM and Aβ is N-methyl-d-aspartate receptor (NMDAR). Glutaminergic neurons are the most vulnerable subtype of neurons in AD, and soluble Aβ enhances glutamate release, leading to excessive activation [120,121]. Whilst NMDAR is critical for synaptic plasticity and neuronal survival, excessive NMDAR activation leads to excitotoxicity and cell death [122]. A recent study in the drosophila model of AD has suggested that overexpression of LAP, the drosophila PICALM orthologue, rescues Aβ42-induced toxicity without affecting Aβ42 level [117].

### 3.3. PICALM and Tau

#### 3.3.1. PICALM Expression and Localization in AD Brains

Different lines of evidence suggest that PICALM expression is dysregulated during the progression of AD. Under a physiological condition in the central nervous system (CNS), PICALM is ubiquitously expressed in various cell types such as endothelial, neuronal and microglial cells [30,32,33,34,35,36,123]. In AD postmortem brains, however, PICALM may undergo an alteration of expression levels and protein localizations in distinct cell types.

During the progression of AD, *PICALM* mRNA expression levels are increased in the brain [32,124] and in the blood [125] of AD patients. In contrast, the PICALM protein level in the total bulk tissue is decreased in AD brains [33]. Another study has shown that PICALM protein level is decreased in the isolated microvessel fraction of AD brains but is not significantly altered in the microvessel-depleted fraction of AD brains [34]. In line with this discordant observation, Tasaki et al., have recently reported that there is a general divergence between the levels of transcripts (mRNA) and protein expression in postmortem AD brain tissue. By mass-spectrometry, they observed a significant reduction of PICALM protein in a correlation with 14.3% faster cognitive decline [124].

Most of the analyses made on PICALM expression were conducted using human postmortem tissues of the affected brain regions that contain different cell types. Single-cell transcriptomic studies have shown that groups of GWAS hit genes, including *PICALM,* are generally upregulated in microglial cells in correlation with neuronal AD pathology [126]. While PICALM immunoreactivity in microglial cells is increased in LOAD brains [33], PICALM immunolabelling is decreased in endothelial cells [30], and PICALM protein level assessed by Western blot is decreased in the microvessel fraction of postmortem AD brain tissues [34]. Neuronal PICALM labeling is increased in AD brains [33,34] and is associated with NFTs (Figure 6A–C) [33]. PICALM and hyperphosphorylated tau are co-immunoprecipitated in AD brain lysate [33]. Since pathological tau in NFTs is ubiquitinated [127], and the ANTH domain of PICALM has an affinity to ubiquitin [67], the interaction between PICALM and tau needs to be further analyzed in non-pathological and pathological conditions to determine the role of ubiquitin to form tau–PICALM complex. Other studies have shown immunoreactivities of PICALM interactors such as PtdIns(4,5)P_2_ [123,128] and AP2A1 [129] in association with phosphotau in NFTs and dystrophic neurites. Of note, PICALM is not the only LOAD GWAS hit protein detected in NFTs: some other proteins encoded by GWAS hits genes are also linked with tau pathologies such as APOE4 [130], CLU [131], BIN1 [132], TPK2B [133] and AP2A1 [129].

#### 3.3.2. PICALM in Primary Tauopathy Brains

To our knowledge, there is no report showing alteration of *PICALM* mRNA expression levels in primary tauopathy brains; however, PICALM protein localization is altered in primary tauopathies with no amyloid pathology [134]. In postmortem human tissues, PICALM immunoreactivity is associated with pathological tau in the AD brain [33] and in other tauopathies such as PSP and Pick disease [134]. In NFTs, PtdIns(4,5)P_2_ and PICALM are colocalized [123,128]. PICALM protein level is decreased in the affected brain areas of various tau-related neurodegenerative diseases, including AD, FTLD-tau-*MAPT*, corticobasal degeneration (CBD) and Lewy body dementia (LBD) diffuse type with tau pathology [33,134,135]. A significant negative correlation was observed between PICALM protein level and hyperphosphorylated tau or autophagy initiation marker Beclin1 in the postmortem human frontal cortex of these neurodegenerative diseases [134]. Such reduction of PICALM seems related to the fact that PICALM is a substrate of both calpain and caspase-3 [33,79,80] that are activated in the brains of AD and other tauopathies [136,137,138,139,140].

#### 3.3.3. PICALM and Tau in Cellular and Animal Models

PICALM modulates tau degradation via both autophagosome formation and autolysosomal fusion in several experimental models, such as cultured cells, drosophila and zebrafish [51]. Knockdown of PICALM caused accelerated accumulation of phosphotau in HeLa cells transiently transfected with DsRed-Tau 4R (four repeats) by impairing autophagy. Nonetheless, they observed in the same study that PICALM overexpression drove tau-mediated neurodegeneration in zebrafish [51]. Thus, they observed that both knockdown and overexpression of PICALM accelerated neurodegeneration in the presence of tau pathology.

Studies in PICALM^+/−^ mice suggest that 50% of PICALM reduction did not trigger a clear motor phenotype or any detectable tau pathology for at least up to 12 months [54,135]. On the contrary, PICALM heteroinsufficiency aggravates tau pathology in a tau transgenic mouse model. By crossing Tg30 overexpressing human double mutant tau (P301S and G272V) transgenic mice developing an age-dependent tau pathology [141] with PICALM^+/−^ mice [54], we generated a novel transgenic mouse line Tg30xPICALM^+/−^ [135]. Tg30xPICALM^+/−^ exhibited an aggravation of motor deficits and tau pathology, suggesting that a reduction of PICALM in the presence of tau pathology may induce deficits in the autophagy pathway [135]. It should be emphasized that this model develops a tau pathology at least partially in a cell-autonomous manner due to a tau transgene bearing double FTLD-related mutations [141]. Since *MAPT* gene mutations are not linked to AD, further studies are required to decipher the role of PICALM in the development and propagation of non-mutant tau.

#### 3.3.4. PICALM and Tau Pathology Propagation

In AD brains, tau pathology develops in a stereotypical manner across neuroanatomically connected networks [142]. Given that tau pathology propagates, at least partially, via clathrin-mediated endocytosis (CME) [143], a clathrin adaptor such as PICALM may be involved in clathrin-mediated tau internalization.

In addition to the prominent role of LRP1 in Aβ clearance [144], recent studies suggest the involvement of LRP1 in regulating tau seeding and internalization [145]. LRP1 is expressed in neuronal, glial and endothelial cells [146]. The affinity of the tau microtubule-binding domain to LRP1 is high under physiological conditions, while hyperphosphorylation of tau leads to a reduction of its binding affinity to LRP1 [147]. Therefore, their interaction may be weaker in AD brains. Since PICALM directly binds to LRP1 to regulate the internalization of LRP1 [34], it can be speculated that PICALM modulates pathological tau spreading via LRP1-mediated pathological tau internalization.

PICALM also modulates cholesterol biogenesis and lipoprotein uptake (see Section 2.1.2). Cholesterol is implicated in amyloid pathology [148] and is detected in amyloid plaques [149]. Since cholesterol depletion promotes tau uptake and aggregation [150], PICALM-mediated misregulation of cholesterol homeostasis may be related to both amyloid and tau pathology progression.

PICALM is involved in iron uptake via transferrin (see Section 2.1.2). Iron in Fe^3+^ (redox-inert state) is detected in NFTs of AD [151] and is implicated in tangle formation [152] and in APP processing [153]. Given PICALM is misregulated in AD brains, the PICALM-mediated iron uptake may be involved in AD pathogenesis.

A study using the HEK tau biosensor cell model [154] has suggested no effect of deletion of GWAS hit genes, including *PICALM,* on the FRET-based signal of tau-tau interaction [155]. Since a lipofectamine-based method was used for PHF internalization, Kolay et al., focused on the effect of a GWAS hit gene depletion on FRET-based tau-tau interaction but not on the effect on the “physiological” uptake of PHF. It remains elusive if PICALM overexpression modulates PHF uptake or tau-tau interaction. Further studies are necessary to assess the role of PICALM in modulating the prion-like tau pathology propagation in in vitro and in vivo models.

### 3.4. PICALM and Glial Cells

Evidence suggests that PICALM immunoreactivity in microglial cells is increased in the postmortem hippocampus of LOAD compared to non-demented controls. Such a striking increase was not evident in human FAD brains (Figure 6D–F) [33]. These observations are in line with recent studies showing the potential involvement of glial cells in AD pathogenesis [156]. Indeed, single cell transcriptome from postmortem AD brains suggests that a group of genes, including *PICALM*, *APOE*, *TREM2*, *MEF2C* and the MHC class II genes *HLA*-*DRB1* and *HLA-DRB5* are upregulated in microglial cells in correlation with AD pathology [126].

*APOE4* carriers exhibit endocytic defects before significant amyloid deposition [157]. *PICALM* is described as a modifier of *APOE4*-induced endocytic defects in human iPSC-derived astrocytes and in yeast. *PICALM* and its yeast orthologue YAP1802 overexpression rescued *APOE4*-induced endosomal abnormalities in human iPSC-derived astrocytes and yeast, respectively [158]. Nonetheless, *PICALM* immunoreactivity is hardly detected in astrocytes in postmortem human brain tissues [33], and thus, the implication of *PICALM* in astrocytic endocytosis might be an adjunct contributor to AD pathogenesis.

Elevated levels of reactive oxygen species (ROS) induce neuronal lipid synthesis that leads to the sequestration of glial lipid droplets (LD) to delay neurotoxicity [159]. *PICALM* modulates this neuroprotective process of neuron-to-glia lipid transport. *PICALM* orthologue LAP knockdown in drosophila glia leads to a significantly reduced LD formation. *PICALM* knockdown in rat astrocytes leads to reduced uptake of neuron-derived fatty acids [159]. PICALM and other proteins encoded by LOAD GWAS hits (APOE, BIN1, CD2AP, AP2A2 and RIN3) are thus involved in LD formation and/or neuron-glial lipid transfer pathway, and knockdown of *PICALM* leads to blunted lipid transfer.

## 4. Discussion

### 4.1. Difficulties in Interpreting Expression Levels of PICALM in AD Brains

Cautions need to be taken to interpret the data about PICALM expression in human autopsy tissues. The results of PICALM expression are often based on the bulk tissue of human postmortem brains [32,33,124]. Since the cell type composition of brain tissue undergoes alterations during the progression of neurodegenerative disease, such bulk data may confound the interpretation of differences in gene expression between AD patients and controls.

In general, there is a widespread heterogeneity between mRNA and protein levels in postmortem human control and AD brains, including *PICALM* [124]. For instance, increased *PICALM* transcripts [32] and decreased PICALM protein level [33,34] was observed in AD brains. Due to such an obvious contradiction of *PICALM* mRNA and protein levels in AD, mRNA data cannot be simply used to predict the protein expression of PICALM. The first possible hypothesis of such divergence is a cellular autoregulatory mechanism with a negative feedback loop [160]. The second possible hypothesis is microRNA (miRNA)-related gene expression regulations. Post-transcriptional repression of mRNA by miRNA has been well-studied in mammalian cells [161]. Studies suggest an important regulatory role of miR-155 in neuroinflammatory conditions, including AD [162]. Interestingly, several independent studies suggest the upregulation of miR-155 in various brain regions of AD patients compared to controls as well as in the brain of transgenic mouse models of Aβ and tau pathologies [163,164]. Indeed, it has been shown that *PICALM* is regulated by miR-155 in mouse brains [165]. Although there is no obvious effect of miRNA-155 knockout on the expression level of *PICALM* transcripts in the mouse brains of these models [166], the effect of miRNA-155 overexpression on *PICALM* (transcript and protein) expression in human AD brains remains elusive. While further study is necessary to better understand the potential regulatory mechanisms of *PICALM* expression, miR-155 may provide a key pathway to untangle the divergence of mRNA and protein levels of *PICALM*. It would be intriguing to find correlation between the PICALM level and miR-155 in further studies. In addition, post-transcriptional regulatory effects of other non-coding RNAs, such as long non-coding RNA and circular RNA on *PICALM,* need to be investigated in AD models. The last but not least hypothesis to explain the divergence of mRNA and protein levels is that PICALM may undergo abnormally accelerated proteolysis in AD brains. Several lines of evidence suggest that PICALM is cleaved by both calpain and caspase [33,79,80], which are activated in AD brains [138]. Moreover, post-translational modifications of PICALM might affect its degradation, stabilization, protein–protein interaction or subcellular localization, yet such effects remain largely elusive. Further studies are thus necessary to decipher the potential abnormalities of PICALM post-translational modifications using 2D Western blot analyses and mass spectrometry from control and AD brain samples to search for AD-specific post-translational modifications of PICALM and their implications in pathology.

### 4.2. Challenges in Deciphering the Effect of PICALM Variants on Its Expression

Several independent studies have recently reported a significant association of the LOAD-susceptibility *PICALM* risk variants (rs10792832 and rs3851179 that are in full LD) on the decreased expression in microglial cells [90,91]. It should be highly informative to create AD models in which microglial PICALM expression is modulated in order to unravel the effect of reduced or increased PICALM expression in microglia.

Additionally, in iPSC-derived endothelial cells, the protective rs3851179^T^ allele is associated with increased mRNA and protein expression of *PICALM* [34] and in bulk tissues when analyzed relative to cell-specific mRNA markers for microvessels and neurons [30]. Given that the effect of rs3851179^T^ is rather modest when normalized to astrocyte marker GFAP (glial fibrillary acidic protein), it becomes more remarkable when normalized to endothelial cell markers or neuronal markers [30]. Therefore, it is possible that *PICALM* variants may have distinct effects on its expression in different cell types.

To decipher the inconsistencies in mRNA and protein levels of PICALM observed in postmortem human AD brain samples [124], the PICALM levels in each cell type (e.g., microglia, neuron, astrocytes, oligodendrocytes, etc.) should be analyzed in large scale diverse cohorts by high throughput single cell sequencing and further validated by RT-qPCR, Western blotting, mass-spectrometry (in distinct cell-type fractions) and by in situ hybridization and/or immunohistochemistry (in sections) of postmortem human AD brain samples [167]. As shown by Zhao et al., on the functional effect of rs3851179 on *PICALM* expression in iPSC-derived endothelial cells, it should be interesting to differentiate iPSC isogenic for rs3851179^T/C^ into various cell types to examine the expression level of PICALM in distinct cell types. More work is necessary to increase the number of samples from different cohorts to improve the power of statistical analyses and pursue proteomic analyses considering heterogeneous cell populations in the brain.

### 4.3. Potential Role of PICALM in Cellular Homeostasis

PICALM is involved in various pathological processes, such as Aβ production and transcytosis, as well as tau pathology progression (Figure 7). Given that PICALM plays versatile roles in iron homeostasis, glutaminergic neurotransmission [117], endocytosis, and autophagy, PICALM may as well be implicated in the “cellular phase” of AD proposed by Bart De Strooper and Eric Karran [168]. In this hypothesis, they propose that the accumulation of amyloid and tau pathologies at the biochemical phase is rather tolerated by neuronal cells. However, the clinical signs manifest only when the cellular homeostatic mechanisms fail along with dysfunction of the neurovascular unit, abnormal neuronal network activity and dyshomeostasis of astrocytes and microglial cells. PICALM and other endocytic proteins encoded by AD susceptibility genes may be implicated in such steps that lead to dysfunction and breakdown of finely tuned cellular physiological functions. PICALM may be a small piece of complex endocytic machinery that may be vulnerable to neurodegeneration [169].

## 5. Conclusions

Multiple lines of evidence suggest that human *PICALM* polymorphism modulates AD pathogenesis. PICALM is involved not only in Aβ production and clearance but also in tau-mediated neurodegeneration and in endocytic homeostasis. The exact role of the polymorphism on *PICALM* expression may vary in different cell types and contribute to multiple pathways in AD pathological mechanisms. Further studies are necessary to decipher the AD-specific post-translational modifications and the functional roles of LOAD-related susceptibility alleles to provide mechanistic insights into the pathological dysregulation of PICALM.

## Figures and Tables

**Figure 1 cells-11-03994-f001:**
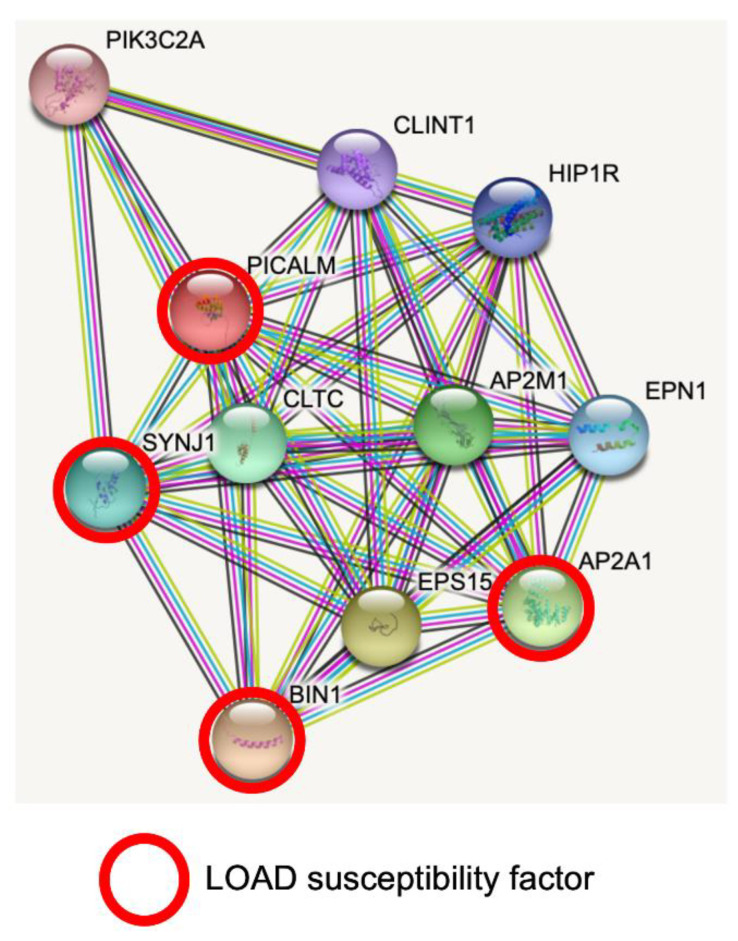
PICALM interacts directly with several GWAS hits identified in LOAD. Predicted protein–protein interactions by STRING database analysis (https://string-db.org (accessed on 1 June 2022)) show several key proteins for clathrin-mediated endocytosis and intracellular trafficking as PICALM interactors [40]. *AP2A1* and *BIN1* have been identified as GWAS hits [41]. SNPs of *SYNJ1* have significant associations with disease onset of AD [45]. CLTC: Clathrin heavy chain 1. SYNJ1: Synaptojanin1. PIK3C2A: Phosphatidylinositol 4-phosphate 3-kinase C2 domain-containing subunit α. CLINT1: Clathrin interactor 1. HIP1R: Huntingtin-interacting protein 1-related protein. AP2M1: AP2 complex subunit mu. EPN1: Epsin-1. AP2A1: AP2 complex subunit α-1. EPS15: Epidermal growth factor receptor substrate 15. BIN1: Myc box-dependent-interacting protein 1.

**Figure 2 cells-11-03994-f002:**
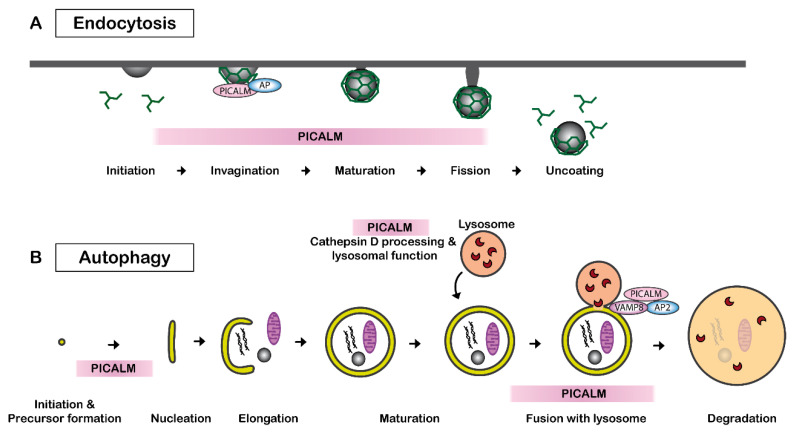
PICALM regulates clathrin-mediated endocytosis and autophagy. (**A**) PICALM is involved in clathrin-mediated endocytosis (CME) as a clathrin adaptor by facilitating the formation of clathrin-coated vesicles (CCVs). The interaction between PICALM, adaptor proteins (AP) and clathrin is critical to maintaining the form and sizes of CCVs. PICALM interacts with AP2 at the plasma membrane and AP 1 in the trans-Golgi network to form CCVs. (**B**) PICALM is also involved in multiple steps of autophagy process. PICALM is involved in the autophagy precursor formation at the autophagosome-lysosome fusion [51], maturation of Cathepsin D and lysosomal function [48].

**Figure 3 cells-11-03994-f003:**
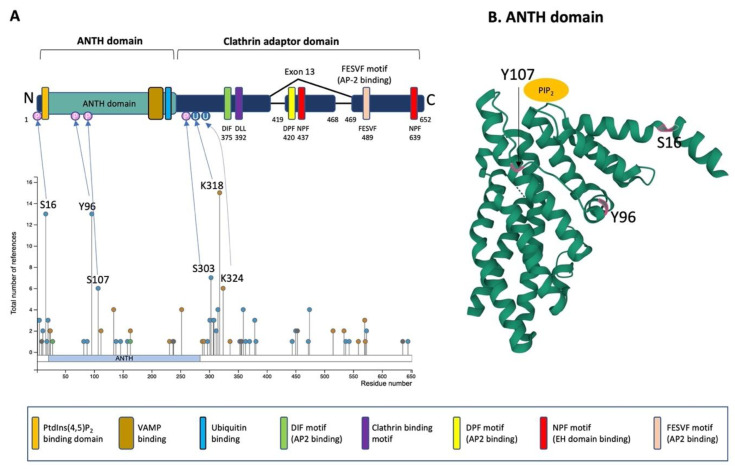
Schematic illustration of PICALM structure and post-translational modifications. (**A**) Schematic structure of PICALM. PICALM is constituted of two distinct domains: the ANTH (AP180 N-terminal homology) domain and clathrin adaptor domain. ANTH domain contains a PtdIns(4,5)P_2_ binding domain and vesicle-associated membrane protein (VAMP) binding domain. Clathrin adaptor domain contains several motifs that are critical to interact with endocytic protein, including DIF, DLL Clathrin binding, DPF, NPF and FESVF motifs. DIF motif binds to AP2. DLL motif is conserved and has weak affinity to Clathrin DPF (Aspartic acid-Proline-Phenylalanine) motifs. NPF (Asparagine-Proline-Phenylalanine) motif binds to AP2 and EH domains (Eps15 Homology domain). Four phosphorylation sites (S16, Y96, S107 and S303) and two ubiquitination sites (K318 and K324) have been described in more than 5 references and validated by mass-spectrometry and thus are highlighted [65]. Altogether, the reported PTMs of PICALM are 41 phosphorylation (S2, S5, T11, S16, T18, S20, S23, T82, Y88, Y96, S107, Y138, T146, T158, Y237, S297, T301, S303, S307, S308, T312, S315, S353, T355, T356, S359, T363, T370, T379, S381, S444, S450, S453, T472, S474, K515, K534, S537, S543, T573 and S645), 15 ubiquitination (K24, K112, K134, K163, K231, K252, K288, K291, K318, K324, K336, K515, K559, K570 and K571), 1 acetylation (K28), 1 mono-methylation (R9), 1 di-methylation (R636) and 1 sumoylation (K238). (**B**) Computational modeling of the structure of PICALM N-terminal ANTH domain is shown (https://www.rcsb.org/structure/3ZYM (accessed on 1 June 2022)) [66]. PICALM ANTH domain is constituted of 11 α-helices. Three known phosphorylation sites (S16, Y96 and S107) in ANTH domain are highlighted in pink.

**Figure 4 cells-11-03994-f004:**
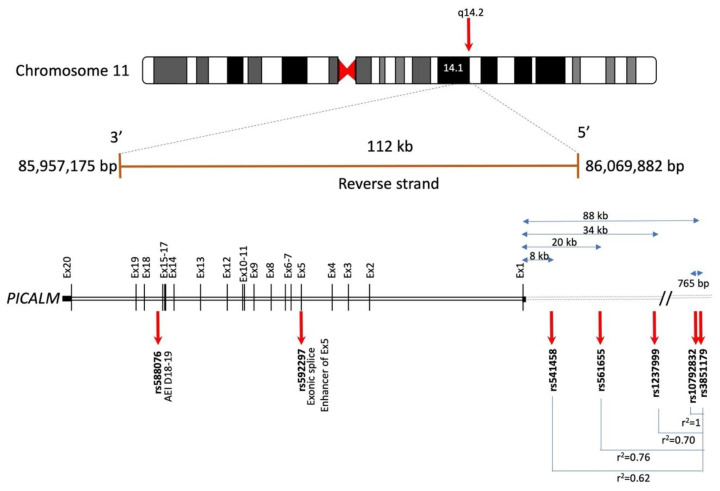
Schematic illustration of *PICALM* genomic location and some of the LOAD-associated SNPs based on the information available in Gene database (https://genome.ucsc.edu/ (accessed on 1 June)) and e!Ensembl (http://www.ensembl.org/ (accessed on 1 June 2022) (accession number ENSG00000073921, GRCh38:CM000673.2). Human *PICALM* gene spans 112 kb on chromosome 11q14 and encodes 20 exons. The most significant LOAD-associated SNP is rs3851179, located approximately 88 kb upstream of *PICALM* in the 5′ intergenic region [17]. rs10792832 is 765 bp from rs3851179 and in full linkage disequilibrium (LD) with rs3851179 (r^2^ = 1). rs1237999, located 34 kb upstream of *PICALM*, is localized in a CCCTC binding factor in a regulatory region [81]. rs561655 is the lead SNP of *PICALM* locus in [82], located 20 kb upstream of *PICALM* gene, a putative transcription factor binding site [15] and in LD with rs3851179 (r^2^ = 0.76). rs541458, located in 8 kb upstream of *PICALM* gene, is also genome-wide significant [15] and is in LD with rs3851179 (r^2^ = 0.62). rs592297 is located in exon 5 and is an exonic splice enhancer [29]. rs588076 is located in the intron 17 of *PICALM* and is associated with the allelic expression imbalance (AEI) in *PICALM* isoform lacking exons 18 and 19 (D18-19) [83].

**Figure 5 cells-11-03994-f005:**
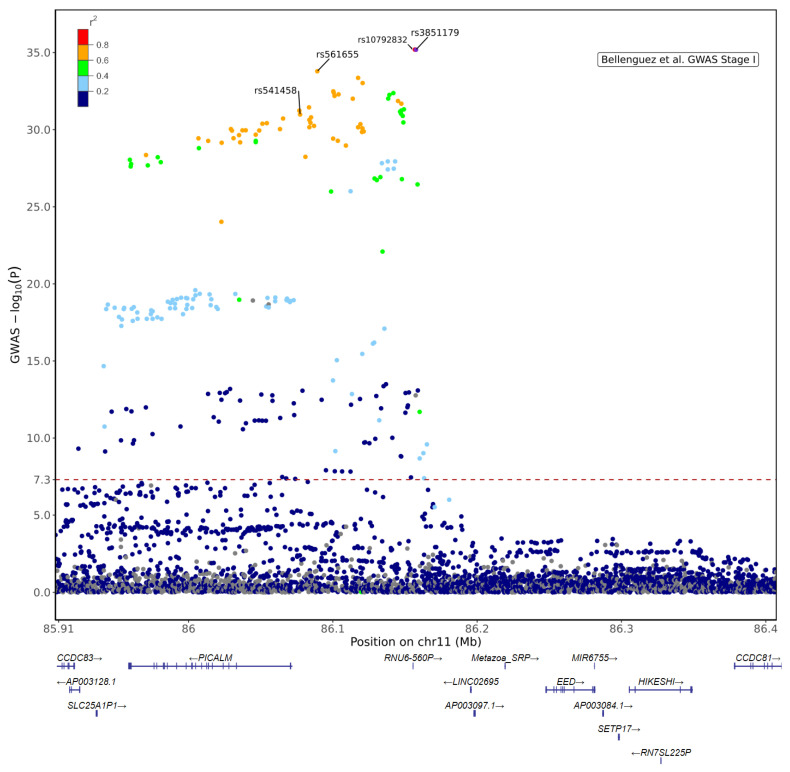
Regional plot for *PICALM* locus based on Stage I results from [17] (Accession No GCST90027158, https://www.ebi.ac.uk/gwas/Study (accessed on 1 July 2022)). The plot shows the lead variant in Stage I + II meta-analysis, rs3851179 and +/− 250 kb extended coordinates for *n* = 3988 variants in chr11:85,907,598–86,407,598 (GRCh38). LD reference data derived from 1000 Genomes Project phased biallelic SNV and INDEL autosomal genetic variants called de novo GRCh38 (for selected *n* = 404 unrelated Non-Finnish European (NFE) sample) [85]. The lead variant rs3851179 is located in the intergenic region between *PICALM* and *EED* in which there are other non-coding genes (*RNU6-560P*, *LINC02695* and *Metazoa SRP*) and a contigs (*AP003097*). rs10792832 (red) is in almost full LD with rs3851179. Other genome-wide significant SNPs, such as rs561655 [82,86] and rs541458 [15,87], are also in high LD with rs3851179. The red dashed line indicates the genome-wide significance threshold (*p* = 5 × 10^−8^).

**Figure 6 cells-11-03994-f006:**
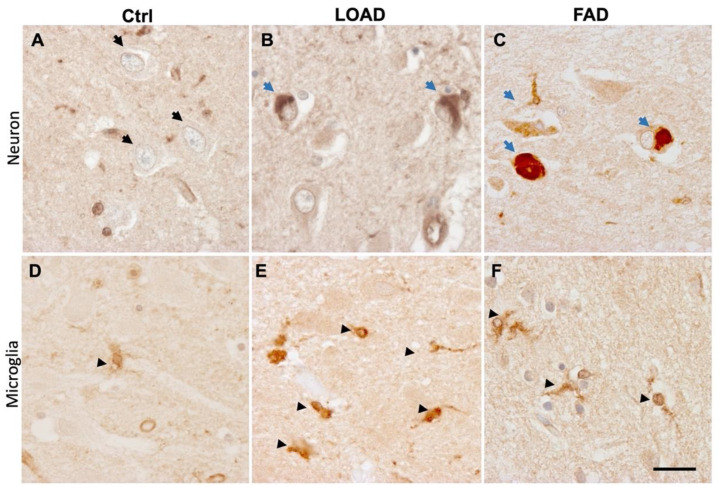
PICALM immunolabelling is increased in LOAD and FAD in correlation with the development of tau pathology. Representative immunolabelling of anti-PICALM antibody (Sigma #HPA019053) in the Cornu Ammonis regions of the hippocampus of control (**A**,**D**), LOAD (**B**,**E**) and FAD with *PSEN1* mutations at R35E and E120D (**C**,**F**). In non-demented control brain, a perinuclear immunolabelling of PICALM was detected in neurons (black arrows) (**A**). A PICALM immunoreactivity was detected in NFTs in neuronal perikarya (blue arrows) in the affected brain areas of LOAD (**B**) and FAD (**C**). PICALM-immunoreactive microglial cells were faintly detected in control brains (arrowheads) (**D**), and the number and the intensity of PICALM-stained microglia were increased in LOAD (**D**) (arrowheads). In FAD (**F**) brains, the increase in PICALM immunostained microglial cells was less prominent than in LOAD. More detailed data can be found in our article [33]. Control (Ctrl), Late-onset Alzheimer’s disease (LOAD), Familial Alzheimer’s disease (FAD). Scale bar 20 µm.

**Figure 7 cells-11-03994-f007:**
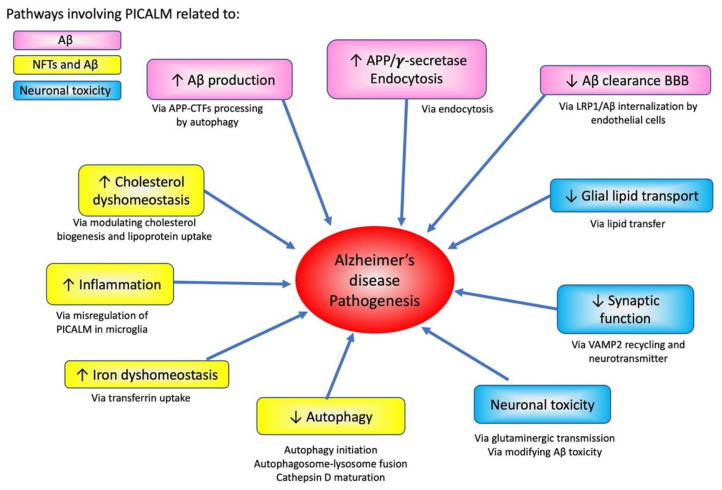
Effects of PICALM on AD pathogenesis pathways and neurodegeneration. PICALM is involved in multiple pathways of AD pathogenesis. PICALM regulates pathways related to APP processing and Aβ-related neurodegeneration (shown in pink). PICALM plays critical role related to both tau and Aβ pathologies, such as cholesterol dyshomeostasis, inflammation, iron dyshomeostasis and autophagy (yellow). PICALM is also crucial for neuronal toxicity via glial lipid transport, synaptic functions, glutaminergic transmission and modifying Aβ-toxicity (blue).

**Table 1 cells-11-03994-t001:** Association between *PICALM* SNPs and AD-related phenotypes. The location of SNPs is based on GRCh38. Linkage disequilibrium to rs3851179 is denoted as r^2^. Effect size shows beta coefficients. SE: standard error. NA: non-available. Hip: hippocampus. EC: entorhinal cortex.

SNP	Location	GWAS p	r^2^	Phenotype	Sample Size	Effect Size (SE)	*p*-Value	Reference
rs510566	85966796	5.669 × 10^−05^	0.042	Risk G allele is associated with lower CSF Aβ42, higher pTau and pTau/Aβ42 ratio	AD: *n* = 114, MCI: *n* = 395, Ctrl: *n* = 203	NA	*p* = 0.048, *p* = 0.0006, *p* = 0.0006	[81]
rs510566	85966796	5.669 × 10^−05^	0.042	Risk G allele is associated with hippocampal atrophy	AD: *n* = 141, MCI: *n* = 461, Ctrl: *n* = 235	NA	*p* < 0.0001	[81]
rs592297	86014894	NA	NA	C allele was associated with faster atrophy rate of hippocampus	AD: *n* = 141, MCI: *n* = 461, Ctrl: *n* = 235	NA	*p* < 0.0045	[81]
rs592297	86014894	NA	NA	C allele was associated with slower atrophy rate of posterior cingulate in MCI	AD *n* = 39, MCI *n* = 422, Ctrl *n* = 257	NA	*p* < 0.05	[97]
rs17148741	86054749	0.07777	0.017	C allele is associated with hippocampal volume	*n* = 1400	NA	*p* = 9.4 × 10^−5^	[96]
rs541458	86077309	1.032 × 10^−31^	0.622	Risk T allele is associated with decreased CSF Aβ	*n* = 412	NA	*p* = 0.002	[98]
rs541458	86077309	1.032 × 10^−31^	0.622	Differential modulation of spontaneous brain activity (C vs. TT)	MCI *n* = 35, Ctrl *n* = 26	NA	*p* < 0.05	[99]
rs541458	86077309	1.032 × 10^−31^	0.622	Risk T allele is associated with an earlier age at midpoint of cognitive decline	*n* = 1831	NA	NA	[100]
rs1237999	86103988	5.091 × 10^−33^	0.696	A allele is associated with lower CSF Aβ42, higher Tau, pTau, pTau/Aβ42 ratio	AD: *n* = 114, MCI: *n* = 395, Ctrl: *n* = 203	NA	*p* = 0.042, *p* = 0.030, *p* = 0.011, *p* = 0.045	[81]
rs543293	86109035	1.07 × 10^−29^	0.637	A allele is associated with atrophy rate of posterior cingulate in MCI	AD *n* = 39, MCI *n* = 422, Ctrl *n* = 257	NA	*p* < 0.05	[97]
rs10501610	86133444	1.186 × 10^−12^	0.082	T allele is associated with slower rise of CSF pTau and pTau/Aβ42 ratio	AD: *n* = 114, MCI: *n* = 395, Ctrl: *n* = 203	NA	*p* = 0.0001	[81]
rs2888903	86143134	1.14 × 10^−28^	0.396	Major allele is associated with earlier age of onset of AD in individuals with down syndrome	*n* = 67	3.31	*p* = 0.011	[101]
rs7110631	86145145	1.367 × 10^−32^	0.740	Risk G allele is associated with an age-dependent cognitive decline	*n* = 1564	NA	*p* = 0.03	[102]
rs7941541	86147496	2.052 × 10^−32^	0.740	Major allele is associated with earlier age of onset of AD in individuals with down syndrome	*n* = 67	Beta: 3.92	*p* = 0.016	[101]
rs10751134	86147845	1.618 × 10^−27^	0.483	Major allele is associated with earlier age of onset of AD in individuals with down syndrome	*n* = 67	Beta: 2.78	*p* = 0.040	[101]
rs3851179	86157598	6.496 × 10^−36^	1	Protective T allele is associated with increased volume of hippocampus and entorhinal cortex	AD *n* = 168, MCI *n* = 357, Ctrl *n*= 215	Beta: Hip: 0.061 (0.029), EC: 0.066 (0.021)	Hip *p* = 0.04, EC *n* = 0.01	[103]
rs3851179	86157598	6.496 × 10^−36^	1	Protective T allele is associated with slower atrophy rate of hippocampus	AD *n* = 205, MCI *n* = 639, Ctrl *n* = 339	NA	*p* < 0.05	[104]
rs3851179	86157598	6.496 × 10^−36^	1	Non-protective C allele is associated with earlier age of onset of AD	*n* = 2569	NA	*p* = 0.0086 (1 side)	[105]
rs3851179	86157598	6.496 × 10^−36^	1	Protective T allele is associated with better cognitive function in the oldest old	*n* = 1369	Beta: 0.60	*p* = 0.024	[106]
rs3851179	86157598	6.496 × 10^−36^	1	Protective T allele is associated with faster information processing speed	*n* = 87	NA	*p* < 0.05	[107]
rs3851179	86157598	6.496 × 10^−36^	1	Protective T allele is associated with larger entorhinal cortical thickness	AD *n* = 245, MCI *n* = 434, Ctrl *n* = 284	Beta: 0.043	*p* = 0.034	[108]

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
