# Peer review of "PICALM and Alzheimer’s Disease: An Update and Perspectives"

_cells, 2022, doi:10.3390/cells11243994_

Round 1

Reviewer 1 Report

Authors elucidate the role of PICALM in AD pathogenesis. This is a well written review.

However, some important references also highlighting the roles of genes and genetic variants in AD pathogenesis have been omitted, example, Karch and Goate, Biol Psychiatry, 2015.

Figure 5 would benefit with more illustration, example, why PICALM is crucial in autophagy, glial lipid transport etc, and how that can affect AD.

With such modifications added, the review should be fine for publication.

Author Response

Reviewer 1

  1. Authors elucidate the role of PICALM in AD pathogenesis. This is a well written review. However, some important references also highlighting the roles of genes and genetic variants in AD pathogenesis have been omitted, example, Karch and Goate, Biol Psychiatry, 2015.

Response from authors

We thank the reviewer for the comment. As suggested, we have added the following important references discussing the roles of genes and gene variants of late-onset Alzheimer’s disease (LOAD)[1, 2].

  1. Figure 5 would benefit with more illustration, example, why PICALM is crucial in autophagy, glial lipid transport etc, and how that can affect AD. With such modifications added, the review should be fine for publication.

Response from authors

We have accordingly modified the Figure 5 (revised Figure 6) with more detailed explanation.

Reviewer 2 Report

In this manuscript, the authors present a comprehensive review of PICALM structure and function, and reports of mutations associated with AD and tauopathies. This is a very difficult topic due to multiple discrepant reports in the literature, which the authors sum up to some degree in the conclusion, with some solid suggestions as to how to remedy the situation. Unfortunately, getting to that point is a very confusing read, and significant re-organization would help. From the beginning, there are issues regarding PICALM localization, one of the topics in the Discussion, but the authors do not reassure the reader that indeed the literature is confusing and just present discrepant results as facts, e.g. a CNS localization sentence that does not mention neurons, followed by a section paragraphs later adding neurons to the mix, using many of the same references. In between, there are in vitro experiments which are described without reference to the cell type. Likewise, the localization of interacting proteins is frequently not described, leaving the reader wondering as to the cellular context in which these interactions are occurring. Some of the localization problems might be clarified with better use of additional single cell RNA sequencing studies that are available, and also spatial transcriptomic studies describing plaque associated genes. Re-organization could include moving all the localization discussion  to a single location in the manuscript, before function and disease are discussed.

The relationship of the PICALM SNPs to structure/function is not adequately discussed, and if the answers are not yet available, that should be better discussed. Do any of the described SNPs affect structure? Where are they located in the gene? How do they affect function?  This discussion should include cholesterol homeostasis and cholesterol-related gene expression, which is becoming more and more prominent in studies of AD pathophysiology, but is not discussed,

The presentation of the literature in regards to neurodegenerative diseases is also unclear, again due to the discrepant results in the literature combined with the localization problems. It is very confusing regarding protective vs risk increase, and this should be presented in a more orderly fashion and better correlated with particular SNPs. Tables would help with some of the required clarifications. The discussion regarding the effects of PICALM level and mutations depending on stage of disease, not just re Abeta processing, is also very important, and something that is becoming an issue for multiple GWAS genes, and deserves a more organized presentation and discussion.

MORE MINOR:

1)      Is FIGURE 4  (PICALM IN LOAD AND FAD) new data? Does it come from a publication and is therefore referenced? Apologies if this reviewer missed that

2)      Better precision is required re gene vs protein, italics vs non-italicized, all caps or first letter caps followed by lower case

3)      Additional publications specifically relating miRNA-155 have been published (Readhead, Haure-Mirande), including publicly available RNA sequencing data, and should be utilized to search if Picalm mRNA slevels are altered

4)      Needs editing re grammar, and here is just one example:  “ It may be also helpful to test isolation of different cell types of brain resident cells in prior to analyses by western blotting, mass-spectrometry, and single cell proteomic [141].”

Author Response

Responses to the editor and the reviewers

Reviewer 2

  1. In this manuscript, the authors present a comprehensive review of PICALM structure and function, and reports of mutations associated with AD and tauopathies. This is a very difficult topic due to multiple discrepant reports in the literature, which the authors sum up to some degree in the conclusion, with some solid suggestions as to how to remedy the situation. Unfortunately, getting to that point is a very confusing read, and significant re-organization would help. From the beginning, there are issues regarding PICALM localization, one of the topics in the Discussion, but the authors do not reassure the reader that indeed the literature is confusing and just present discrepant results as facts, e.g. a CNS localization sentence that does not mention neurons, followed by a section paragraphs later adding neurons to the mix, using many of the same references.

Response from authors

We have changed the title of the section 2.1 to “Cellular function and tissue expression of PICALM”. We apologize for not clearly mentioning the neuronal expression of PICALM whereas it has been described in several independent studies [1-4].We have accordingly added this information in section 2.1.

  1. In between, there are in vitro experiments which are described without reference to the cell type. Likewise, the localization of interacting proteins is frequently not described, leaving the reader wondering as to the cellular context in which these interactions are occurring.

Response from authors

We have clarified the cell types in the sections 2.1.1-2.1.4 as highlighted in the revised manuscript.

  1. Some of the localization problems might be clarified with better use of additional single cell RNA sequencing studies that are available, and also spatial transcriptomic studies describing plaque associated genes.

Response from authors

We have searched for articles and data banks of the spatial transcriptomic analyses [5] and found that Picalm is highly expressed in wild-type mouse hippocampus granule cells via data bank of ALZMAP [6]. This information has been added in the section 2.1.

  1. Re-organization could include moving all the localization discussion  to a single location in the manuscript, before function and disease are discussed.

Response from authors

We agree with the reviewer in that we need to improve and simplify the structure. Unfortunately, we were not able to make a single chapter on the localization of PICALM because chapter 2 is dedicated to the general and physiological (non-pathological) tissue expression and functions of PICALM while chapter 3 discuss the AD-related discoveries on PICALM. We needed to maintain the original structure in which we can highlight the disease-related alteration of PICALM localization.

  1. The relationship of the PICALM SNPs to structure/function is not adequately discussed, and if the answers are not yet available, that should be better discussed. Do any of the described SNPs affect structure? Where are they located in the gene? How do they affect function?  

Response from authors

There are no LOAD-associated coding variants of PICALM found [7]. Therefore, it is presumed that there is no effect of LOAD-related SNPs on the protein structure of PICALM. However, the PICALM expression level is affected by the LOAD-associated SNP rs3851179 [2] and is discussed in the section 3.1.

  1. This discussion should include cholesterol homeostasis and cholesterol-related gene expression, which is becoming more and more prominent in studies of AD pathophysiology, but is not discussed.

Response from authors

We agree with the reviewer and have accordingly added a paragraph to discuss the recent discoveries regarding cholesterol dyshomeostasis in AD brains and the implication in AD pathogenesis in 3.3.4.

  1. The presentation of the literature in regards to neurodegenerative diseases is also unclear, again due to the discrepant results in the literature combined with the localization problems. It is very confusing regarding protective vs risk increase, and this should be presented in a more orderly fashion and better correlated with particular SNPs. Tables would help with some of the required clarifications. The discussion regarding the effects of PICALM level and mutations depending on stage of disease, not just re Abeta processing, is also very important, and something that is becoming an issue for multiple GWAS genes, and deserves a more organized presentation and discussion.

Response from authors

We have added a table on the summary and the observations related to the SNPs susceptible for LOAD (Table 1 and 2) in order to clarify the points suggested.

MORE MINOR:

  • Is FIGURE 4  (PICALM IN LOAD AND FAD) new data? Does it come from a publication and is therefore referenced? Apologies if this reviewer missed that

Response from authors

These data can be found in our previously published article [1]. The photos used in this article are not the identical photos, thus there is no conflict of copyright. We have added a phrase to explain that this observation has been previously described in the figure legend (revised figure 5).

  • Better precision is required re gene vs protein, italics vs non-italicized, all caps or first letter caps followed by lower case

Response from authors

We have checked all the characters throughout the manuscript.

  • Additional publications specifically relating miRNA-155 have been published (Readhead, Haure-Mirande), including publicly available RNA sequencing data, and should be utilized to search if PicalmmRNA levels are altered

Response from authors

We would like to sincerely thank the reviewer for having suggested to look for further information on miRNA-155. In the mentioned article [8]and available data from its gene expression data (supplementary material), there was no significant effect of miRNA-155 deletion on the mRNA expression of Picalm in the comparison of the following models (wild-type, mir155KO, APPPS1 and APPPS1_mirKO).

­­­

ensembl

symbol

log2FoldChange

pvalue

mirKO_vs_WT_8mo

ENSMUSG00000039361

Picalm

0,047807

0,381610228

APPPS1_mirKO_vs_APPPS1_8mo

ENSMUSG00000039361

Picalm

-0,0239

0,618888145

APPPS1_vs_WT_8mo

ENSMUSG00000039361

Picalm

0,018773

0,71875159

However, this does not eliminate the possibility that mi-RNA155 upregulation observed in human AD brains would alter the level of PICALM transcript. This discussion has been added in the revised manuscript (Chapter 4.1).

4)      Needs editing re grammar, and here is just one example:  “ It may be also helpful to test isolation of different cell types of brain resident cells in prior to analyses by western blotting, mass-spectrometry, and single cell proteomic [141].”

Response from authors

The manuscript has been thoroughly checked by a native English speaker.

Refeernces

  1. Ando, K.; J. P. Brion; V. Stygelbout; V. Suain; M. Authelet; R. Dedecker; A. Chanut; P. Lacor; J. Lavaur; V. Sazdovitch; et al. Clathrin Adaptor Calm/Picalm Is Associated with Neurofibrillary Tangles and Is Cleaved in Alzheimer's Brains. Acta Neuropathol. 2013, 125, 861-78.
  2. Zhao, Z.; A. P. Sagare; Q. Ma; M. R. Halliday; P. Kong; K. Kisler; E. A. Winkler; A. Ramanathan; T. Kanekiyo; G. Bu; et al. Central Role for Picalm in Amyloid-Beta Blood-Brain Barrier Transcytosis and Clearance. Nat Neurosci. 2015, 18, 978-87.
  3. Xiao, Q.; S. C. Gil; P. Yan; Y. Wang; S. Han; E. Gonzales; R. Perez; J. R. Cirrito; J. M. Lee. Role of Phosphatidylinositol Clathrin Assembly Lymphoid-Myeloid Leukemia (Picalm) in Intracellular Amyloid Precursor Protein (App) Processing and Amyloid Plaque Pathogenesis. J Biol Chem. 2012, 287, 21279-89.
  4. Petralia, R. S.; P. J. Yao. Ap180 and Calm in the Developing Hippocampus: Expression at the Nascent Synapse and Localization to Trafficking Organelles. J Comp Neurol. 2007, 504, 314-27.
  5. Chen, W. T.; A. Lu; K. Craessaerts; B. Pavie; C. Sala Frigerio; N. Corthout; X. Qian; J. Lalakova; M. Kuhnemund; I. Voytyuk; et al. Spatial Transcriptomics and in Situ Sequencing to Study Alzheimer's Disease. Cell. 2020, 182, 976-91 e19.
  6. ———. "Alzmap." https://alzmap.org/ (accessed April 29 2018).
  7. Schnetz-Boutaud, N. C.; J. Hoffman; J. E. Coe; D. G. Murdock; M. A. Pericak-Vance; J. L. Haines. Identification and Confirmation of an Exonic Splicing Enhancer Variation in Exon 5 of the Alzheimer Disease Associated Picalm Gene. Ann Hum Genet. 2012, 76, 448-53.
  8. Readhead, B.; J. V. Haure-Mirande; D. Mastroeni; M. Audrain; T. Fanutza; S. H. Kim; R. D. Blitzer; S. Gandy; J. T. Dudley; M. E. Ehrlich. Mir155 Regulation of Behavior, Neuropathology, and Cortical Transcriptomics in Alzheimer's Disease. Acta Neuropathol. 2020, 140, 295-315.

Reviewer 3 Report

The authors thoroughly reviewed the PICALM literature and the evidence for its role in Alzheimer's disease (AD). That's a tremendous amount of work! However, I feel that this m.s. is a long and encylopedia-like article with a very narrow focus on PICALM as one of the many AD genes . This may not be suitable for the readers of Cells, but more for a more specialised journal.

My main expertise lies in the genetics and epidemiology of AD. Therefore, I cannot assess the review of the biochemical considerations or functional in vitro and in vivo studies. However, for the genetics part, I am missing a bit the post-GWAS functional annotation of the top associated variants. Molecular eQTL effects, CADD scores, regulatory features? Furthermore, are there any SNPs in LD with the top associated one that are functionally more interesting? Are they all in the PICALM gene region or are there also other SNPs in LD? 

Other comments:

- There is no mentioning of LD of the listed variants - are they independent from each other, or do they represent the same GWAS signal? Are there secondary signals coming from conditional analyses?

- What's the allele frequency of the top associated variant (it should not be the "most validated" SNP, but the most significant SNP based on recent GWAS)? Is there evidence for an association of rare variants in PICALM with AD? 

- Also, in reporting SNP associations, please stick to more stringent p values than nominal significance to report association results. The best would be to use results from recent large GWAS. For instance, as far as I know, PICALM is NOT associated with Parkinson's disease judging by recent GWAS.

Author Response

Reviewer 3

  • The authors thoroughly reviewed the PICALM literature and the evidence for its role in Alzheimer's disease (AD). That's a tremendous amount of work! However, I feel that this m.s. is a long and encyclopedia-like article with a very narrow focus on PICALM as one of the many AD genes . This may not be suitable for the readers of Cells, but more for a more specialised journal. My main expertise lies in the genetics and epidemiology of AD. Therefore, I cannot assess the review of the biochemical considerations or functional in vitro and in vivo studies. However, for the genetics part, I am missing a bit the post-GWAS functional annotation of the top associated variants.

Response from authors

We thank the reviewer for the comments. As suggested, we have added some critical information on the post-GWAS functional annotation of the top associated variants in Chapter 1.

  • Molecular eQTL effects, CADD scores, regulatory features?

Response from authors

We have accordingly added information regarding the molecular eQTL effects and regulatory features in the chapter 3.1.

Regarding the combined annotation dependent depletion (CADD) scores, computational analyses by PredictSNP2 predict five potentially deleterious synonymous SNPs of PICALM [1]. However, none of LOAD-associated SNPs of PICALM (listed in [2]) had a significantly high CADD score (discussed in the revised Chapter 3.1).

  • Furthermore, are there any SNPs in LD with the top associated one that are functionally more interesting? Are they all in the PICALM gene region or are there also other SNPs in LD? There is no mentioning of LD of the listed variants - are they independent from each other, or do they represent the same GWAS signal?

Response from authors

The most significantly LOAD-associated SNP is rs3851179, located 88 kb upstream of PICALM gene. rs3851179 is adjacent to transcription factor binding region and associated with expression of PICALM mRNA in endothelial cells [3] and in the brain [2] notably in the frontal and temporal cortex [4]. Therefore, we consider that rs3851179 could be a functional variant that regulates PICALM expression. Furthermore, rs3851179 is in linkage disequilibrium (LD) with several SNPs such as rs541458 [5], rs592297, an exonic splice enhancer in exon 5, [6], rs642949 [7], rs1237999 and rs7114401 [2]. We have revised chapter 3.1 to clearly discuss this information.

  • Are there secondary signals coming from conditional analyses?

In order to answer this question, we looked for a recent study on fine mapping, in which the authors identified credibly causal variants. They show that the several LOAD-associated PICALM variants provided only one single signal, suggesting that there should be only a single causal variant of PICALM (please refer to the Figure 3a of  [8]). However, it remains elusive if this study was not powerful enough to identify more causal signals. Therefore, more studies are necessary to respond this very interesting question.

  • What's the allele frequency of the top associated variant (it should not be the "most validated" SNP, but the most significant SNP based on recent GWAS)?

Response from authors

We agree that the allele frequency needs to be shown for top associated SNPs. We have accordingly made a table of the SNPs of PICALM associated with LOAD with the information on the minor allele frequency (MAF) (revised Table 1). As suggested, “most validated SNP” has been changed to “most significant SNP” in the revised manuscript.

  • Is there evidence for an association of rare variants in PICALM with AD? 

Response from authors

Two rare nonsynonymous coding variants in PICALM have been reported [9]. These two rare variants, rs147556602 (p.P495A) and rs117411388 (p.H458R) of PICALM, changes the amino acids in the clathrin-adaptor domain of PICALM protein. However, no association between these rare variants and LOAD susceptibility has been described so far and further study is necessary. This information has been included in section 3.1.

  • Also, in reporting SNP associations, please stick to more stringent p values than nominal significance to report association results. The best would be to use results from recent large GWAS. For instance, as far as I know, PICALM is NOT associated with Parkinson's disease judging by recent GWAS.

Response from authors

We agree with the reviewer. We have accordingly deleted the sentence regarding the potential implication of PICALM SNP in Parkinson’s disease and metabolic syndrome since the p values from the mentioned articles do not reach the statistical significance thresholds of GWAS (p<5x10-8) [10].

References:

  1. Tey, H. J.; C. H. Ng. Computational Analysis of Functional Snps in Alzheimer's Disease-Associated Endocytosis Genes. PeerJ. 2019, 7, e7667.
  2. Xu, W.; C. C. Tan; X. P. Cao; L. Tan; Initiative Alzheimer's Disease Neuroimaging. Association of Alzheimer's Disease Risk Variants on the Picalm Gene with Picalm Expression, Core Biomarkers, and Feature Neurodegeneration. Aging (Albany NY). 2020, 12, 21202-19.
  3. Zhao, Z.; A. P. Sagare; Q. Ma; M. R. Halliday; P. Kong; K. Kisler; E. A. Winkler; A. Ramanathan; T. Kanekiyo; G. Bu; et al. Central Role for Picalm in Amyloid-Beta Blood-Brain Barrier Transcytosis and Clearance. Nat Neurosci. 2015, 18, 978-87.
  4. Thomas, R. S.; A. Henson; A. Gerrish; L. Jones; J. Williams; E. J. Kidd. Decreasing the Expression of Picalm Reduces Endocytosis and the Activity of Beta-Secretase: Implications for Alzheimer's Disease. BMC Neurosci. 2016, 17, 50.
  5. Harold, D.; R. Abraham; P. Hollingworth; R. Sims; A. Gerrish; M. L. Hamshere; J. S. Pahwa; V. Moskvina; K. Dowzell; A. Williams; et al. Genome-Wide Association Study Identifies Variants at Clu and Picalm Associated with Alzheimer's Disease. Nat Genet. 2009, 41, 1088-93.
  6. Schnetz-Boutaud, N. C.; J. Hoffman; J. E. Coe; D. G. Murdock; M. A. Pericak-Vance; J. L. Haines. Identification and Confirmation of an Exonic Splicing Enhancer Variation in Exon 5 of the Alzheimer Disease Associated Picalm Gene. Ann Hum Genet. 2012, 76, 448-53.
  7. Furney, S. J.; A. Simmons; G. Breen; I. Pedroso; K. Lunnon; P. Proitsi; A. Hodges; J. Powell; L. O. Wahlund; I. Kloszewska; et al. Genome-Wide Association with Mri Atrophy Measures as a Quantitative Trait Locus for Alzheimer's Disease. Mol Psychiatry. 2011, 16, 1130-8.
  8. Schwartzentruber, J.; S. Cooper; J. Z. Liu; I. Barrio-Hernandez; E. Bello; N. Kumasaka; A. M. H. Young; R. J. M. Franklin; T. Johnson; K. Estrada; et al. Genome-Wide Meta-Analysis, Fine-Mapping and Integrative Prioritization Implicate New Alzheimer's Disease Risk Genes. Nat Genet. 2021, 53, 392-402.
  9. Vardarajan, B. N.; M. Ghani; A. Kahn; S. Sheikh; C. Sato; S. Barral; J. H. Lee; R. Cheng; C. Reitz; R. Lantigua; et al. Rare Coding Mutations Identified by Sequencing of Alzheimer Disease Genome-Wide Association Studies Loci. Ann Neurol. 2015, 78, 487-98.
  10. Chen, Z.; M. Boehnke; X. Wen; B. Mukherjee. Revisiting the Genome-Wide Significance Threshold for Common Variant Gwas. G3 (Bethesda). 2021, 11.

Round 2

Reviewer 3 Report

The authors have considered my comments. This has somewhat improved the m.s. However, the authors still only provide a very incomplete genetic picture of this one single locus. The authors may want to consider to involve an author with a more extensive expertise and experience in AD GWAS methodology to improve this review in terms of the presentation of the genetic findings and implications.

The newly inserted table is not very helpful and can be removed as is. Crucial systematic LD information is still missing. The authors should focus on the top SNP from the latest 1-2 GWAS only (Bellenguez, Whitman) and evaluate the LD of the surrounding SNPs (e.g., +/- 200kb) and their functional implications (molecular consequence, functional annotations) in an unbiased fashion based on current genomic databases (without referencing other studies). This can be then summarized in a table or LD plot and presented/interpreted in the text. 

In line with this, it needs to be addressed whether PICALM is the only plausible candidate gene in this locus.

I'd also would like to see a table providing an overview of the other genome-wide genetic findings for PICALM that are now only described in the text including the following information: study author and publication year,  outcome phenotype, sample size, top PICALM SNP (LD to top AD risk SNP from Bellenguez or Whitman), effect allele, effect size (if appropriate), p value.

Author Response

We would like to sincerely thank you for your critics. 

Comments from Reviewer #3 (2nd round)

  1. "The authors have considered my comments. This has somewhat improved the m.s. However, the authors still only provide a very incomplete genetic picture of this one single locus. The authors may want to consider to involve an author with a more extensive expertise and experience in AD GWAS methodology to improve this review in terms of the presentation of the genetic findings and implications".

Response from the authors:

We have now involved Dr. Fahri Küçükali (GWAS expert) as collaborator and co-author to add the suggested genetic analyses to improve the manuscript.

  1. "The newly inserted table is not very helpful and can be removed as is. Crucial systematic LD information is still missing".

Response from the authors:

The previous table 1 has been accordingly deleted.

  1. "The authors should focus on the top SNP from the latest 1-2 GWAS only (Bellenguez, Whitman) and evaluate the LD of the surrounding SNPs (e.g., +/- 200kb) and their functional implications (molecular consequence, functional annotations) in an unbiased fashion based on current genomic databases (without referencing other studies). This can be then summarized in a table or LD plot and presented/interpreted in the text”.

Response from the authors:

The top SNPs from the suggested articles are rs3851179 (Bellenguez et al., 2022) and rs561655 (Wightman et al., 2021). However, for Wightman et al. the GWAS summary statistics file does not include a substantial part of their GWAS (23andme), and they do not provide OR and 95% CI in their results, making the interpretation of this GWAS more difficult. Additionally, when 23andme samples are excluded, rs3851179 was more significant than rs561655 in GWAS of Wightman et al. 

We have thus concentrated on the lead GWAS hit rs3851179 (Bellenguez et al., 2022) and have added the information on the LD data of rs3851179  in the surrounding region (+/-250 kb) and listed their functional implications in the revised Figure 5. The exact r2 values are shown in the supplementary Table 1 that summarizes GWAS significant SNPs of PICALM locus.

  1. "In line with this, it needs to be addressed whether PICALM is the only plausible candidate gene in this locus".

Response from the authors:

Since recent eQTL studies have provided substantial evidence on a significant colocalization of PICALM GWAS signal and expression in microglia [1-4] via chromatin accessibility [2]. These studies support the hypothesis that PICALM is likely to be the sole prioritized and causal risk gene in the locus. This discussion has been added to the section 3.1.2.

  1. "I'd also would like to see a table providing an overview of the other genome-wide genetic findings for PICALM that are now only described in the text including the following information: study author and publication year, outcome phenotype, sample size, top PICALM SNP (LD to top AD risk SNP from Bellenguez or Whitman), effect allele, effect size (if appropriate), p value."

Response from the authors:

As suggested, we have added a table with the requested information in the revised manuscript (revised Table 1).

References:

  1. Bryois, J.; D. Calini; W. Macnair; L. Foo; E. Urich; W. Ortmann; V. A. Iglesias; S. Selvaraj; E. Nutma; M. Marzin; et al. Cell-Type-Specific Cis-Eqtls in Eight Human Brain Cell Types Identify Novel Risk Genes for Psychiatric and Neurological Disorders. Nat Neurosci. 2022, 25, 1104-12.
  2. Kosoy, R.; J. F. Fullard; B. Zeng; J. Bendl; P. Dong; S. Rahman; S. P. Kleopoulos; Z. Shao; K. Girdhar; J. Humphrey; et al. Genetics of the Human Microglia Regulome Refines Alzheimer's Disease Risk Loci. Nat Genet. 2022, 54, 1145-54.
  3. Young, A. M. H.; N. Kumasaka; F. Calvert; T. R. Hammond; A. Knights; N. Panousis; J. S. Park; J. Schwartzentruber; J. Liu; K. Kundu; et al. A Map of Transcriptional Heterogeneity and Regulatory Variation in Human Microglia. Nat Genet. 2021, 53, 861-68.
  4. Lopes, K. P.; G. J. L. Snijders; J. Humphrey; A. Allan; M. A. M. Sneeboer; E. Navarro; B. M. Schilder; R. A. Vialle; M. Parks; R. Missall; et al. Genetic Analysis of the Human Microglial Transcriptome across Brain Regions, Aging and Disease Pathologies. Nat Genet. 2022, 54, 4-17.